# Enhanced Energy Storage Performance of AgNbO$_3$:$x$CeO$_2$ by Synergistic Strategies of Tolerance Factor and Density Regulations

Ke An [1], Gang Li [2,*], Tingting Fan [1], Feng Huang [1], Wenlin Wang [1] and Jing Wang [1,*]

1    Key Laboratory of Analytical Science and Technology of Hebei Province,
College of Chemistry and Materials Science, Hebei University, Baoding 071002, China
2    Department, Baoding Green Yijia Environmental Protection Technology Ltd., Baoding 071002, China
*    Correspondence: lyj17631583999@163.com (G.L.); wangjing9804@163.com or wangjing@hbu.edu.cn (J.W.)

**Abstract:** AgNbO$_3$-based ceramics have been widely studied as ideal lead-free materials. Herein, AgNbO$_3$:$x$CeO$_2$ ($x$ = 0, 1, 2 mol%) ceramics were successfully prepared by the conventional solid-state reaction method. The optimization of energy storage properties is ascribed to the enhanced antiferroelectric (AFE) stability and the increased breakdown strength ($E_b$). The reduction of the tolerance factor leads to the enhancement of AFE stability. In addition, the enhancement of $E_b$ is due to the increase of actual density, which is achieved through the regulation of CeO$_2$ amount and grinding procedure in the experimental process. A high recoverable energy density ($W_{rec}$) of 5.04 J/cm$^3$ and an energy efficiency ($\eta$) of 46.2% were achieved in AgNbO$_3$:0.01CeO$_2$ ceramics under an applied electric field up to 390 kV/cm. A higher $\eta$ of 55.4% was obtained in AgNbO$_3$:0.02CeO$_2$ components. This research provides guidance for finding ceramic materials with comprehensive energy storage properties.

**Keywords:** antiferroelectric materials; energy storage performance; AgNbO$_3$; breakdown strength; grain size heterogeneity

## 1. Introduction

Subject to the energy crisis and environmental problems caused by economic development, people have begun to vigorously develop sustainable energy [1,2]. Among various measures, addressing the issue of electrical energy storage is the key to efficiently using energy. Batteries, electrochemical capacitors, and dielectric capacitors are commonly used energy storage devices today [3–5]. Compared with other devices, dielectric capacitors have the characteristics of higher power density, fast charge and discharge capability, and long cycle life. However, the low energy density and energy efficiency of dielectric ceramics limit its application in many fields [5–7].

Theoretically, the energy storage performance can be calculated for dielectrics by the following equations [8].

$$W = \int_0^{P_{max}} E\,dP \qquad (1)$$

$$W_{rec} = -\int_{P_{max}}^{P_r} E\,dP \qquad (2)$$

$$\eta = \frac{W_{rec}}{W} \times 100\% = \frac{W_{rec}}{W_{rec} + W_{loss}} \times 100\% \qquad (3)$$

where $W$, $W_{rec}$, $W_{loss}$, $\eta$, $P_r$, $P_{max}$, and $E$ represent energy storage density, recoverable energy density, energy loss density, energy storage efficiency, remanent polarization, maximum polarization, and applied electric field, respectively,. It is clear that the key to obtaining dielectric energy storage materials with high performance is to achieve high

$P_{max}$, low $P_r$, and high dielectric breakdown strength ($E_b$). Compared with linear dielectric materials and ferroelectric (FE) materials, antiferroelectric (AFE) materials with relatively high energy storage density have become a research hotspot in recent years [9–11]. The earliest discovered AFE material was lead zirconate (PbZrO$_3$), followed by the discovery of Pb(Zr$_x$Ti$_{1-x}$)O$_3$ (PZT) materials with specific compositions and PbHfO$_3$-based ceramics. Although lead-based materials currently provide $W_{rec}$ up to 11 J/cm$^3$, the use of lead poses an environmental hazard. Hence lead-based materials are gradually abandoned [12,13]. Silver niobate AFE material is expected to become a strong competitor to replace lead-based materials. AgNbO$_3$ is a material with abundant phase transitions at elevated temperatures, including M$_1$, M$_2$, M$_3$, O, T, and C phases. Among them, the O (orthorhombic), T (tetragonal), and C (cubic) phases are paraelectric. M$_2$ and M$_3$ are antiferroelectric phases, and M$_1$ is a ferrielectric phase. The phase transition between M phases is mainly related to cation displacement, while the transition among M, O, T, and C phases is related to oxygen octahedral tilting [14,15].

In recent years, AgNbO$_3$ (AN) based ceramic capacitors have been extensively studied due to their high $P_{max}$ and greenness. However, many properties of pure AgNbO$_3$ limit its energy storage density and energy storage efficiencies, such as weak ferroelectricity at room temperature, low FE-AFE phase transition electric field ($E_A$), large hysteresis ($\Delta E$), and low breakdown strength ($E_b$) [16,17]. Moreover, large $P_r$ will also limit its energy storage efficiency. Thus, it is of vital importance to stabilize the antiferroelectric (AFE) phase of AgNbO$_3$. Goldschmidt tolerance factor ($t$) is a key index to assess the phase stability of the perovskite structure.

$$t = (R_A + R_B)/\sqrt{2}(R_B + R_O) \tag{4}$$

where $R_A$, $R_B$, and $R_O$ represent the radii of A-site ions, B-site ions, and oxygen ions, respectively. When $t > 1$, the ferroelectric phase is more stable, while when $t < 1$, the antiferroelectric phase is more stable [18–21]. In order to obtain a decreased t, doping modification is a common method. The introduction of smaller radius ions at the A-site and larger radius ions at the B-site can effectively reduce $t$, thereby enhancing the AFE stability. Song et al. [18] introduced BiMnO$_3$ into AgNbO$_3$ and obtained a $W_{rec}$ of 2.4 J/cm$^3$ under the influence of enhanced AFE stability. Han et al. [19] doped Sr$^{2+}$ into AgNbO$_3$ and obtained a $W_{rec}$ of 2.9 J/cm$^3$ at a low applied electric field of 190 kV/cm. As an aliovalent doping ion, it will induce A-site vacancies, and its presence will refine the grain size to achieve a higher breakdown strength. Later, the doping of lanthanide elements such as Gd$^{3+}$ [20] and Sm$^{3+}$ [21] further increased the breakdown strength. In addition, the phase transition field of $E_A$ enhances with the increase of doping concentration owing to the enhanced AFE stability. The combination of the two factors resulted in $W_{rec}$ of 4.5 J/cm$^3$ and 5.2 J/cm$^3$, respectively. In addition to adjusting the tolerance factor, Zhao et al. [8] reported the reduction of B-site cation polarizability also contributes to enhanced AFE stability. In the same year, they prepared AgNbO$_3$-0.1wt%WO$_3$ and achieved the $W_{rec}$ of 3.3 J/cm$^3$ [22]. The co-doping of A-site ions and B-site ions can also effectively optimize energy storage performance. Shang et al. [23] constructed Ag$_{0.97}$Nd$_{0.015}$Nb$_{0.985}$Hf$_{0.015}$O$_3$ ceramic and realized a $W_{rec}$ of 3.94 J/cm$^3$ under 235 kV/cm. Han et al. [24] introduced Sm$^{3+}$ (A-site) and Ta$^{5+}$ (B-site) into AgNbO$_3$ simultaneously and obtained a $W_{rec}$ of 4.87 J/cm$^3$.

Defect engineering is thought to be an effective method to optimize energy storage performance. The enhanced properties can be attributed to lattice distortion caused by doping ions with unequal radii into the materials. Zhang et al. [25] doped Sr$^{2+}$ into (Bi$_{0.5}$Na$_{0.5}$)TiO$_3$ based on A-site vacancy engineering and introduced Sr$_{0.85}$Bi$_{0.1}$ZrO$_3$ (SBZ) into (Bi$_{0.5}$Na$_{0.5}$)$_{0.7}$Sr$_{0.3}$TiO$_3$ (BNST). The presence of Sr$^{2+}$ effectively suppresses $P_r$ and significantly strengthens relaxation characteristics, and thus a high $W_{rec}$ of 3.53 J/cm$^3$ and a high $\eta$ of 87.15% was obtained in the ceramic. Aliovalent A-site engineering has also been widely applied in AgNbO$_3$-based ceramics. Relevant studies have shown that A-site vacancies due to the aliovalent doping are conducive to enhancing polarization. Luo et al. [26] introduced Ca$^{2+}$ into AgNbO$_3$ and found that Ca$^{2+}$ doping can produce A-site

vacancy. The polarizability and dielectric constant increase monotonically with increasing $Ca^{2+}$ from 1 mol% to 4 mol%. This is similar to "soft" doping in ferroelectric materials. The $P_{max}$ was enhanced to 39.6 $\mu C/cm^2$, and a $W_{rec}$ of 3.55 $J/cm^3$ under 220 kV/cm was obtained in $Ag_{0.92}Ca_{0.04}NbO_3$ ceramics. In addition, they reached the same conclusion for $La^{3+}$-doped ceramics, and a $W_{rec}$ of 3.12 $J/cm^3$ was achieved in $Ag_{0.94}La_{0.02}NbO_3$ ceramics [27].

Moreover, the enhancement of the $E_b$ also plays a vital role in the enhancement of the energy storage density. $E_b$ is an important parameter for analyzing energy storage performance, which is affected by factors such as porosity, grain size, and defects [28,29]. Theoretically, ceramics with dense microstructure are more accessible to obtain higher $E_b$ [30]. This is attributed to the fact that the gas existing in voids possesses a low dielectric permittivity, which causes the voids to need to bear a higher local electric field. However, with the increase of the applied electric field, a local breakdown can easily occur due to the lower $E_b$ of gas [4]. The commonly used strategies are liquid phase sintering and pressure-assisted sintering for reducing porosity and increasing density. Xu et al. [31] added $BaCu(B_2O_5)$ (BCB) with a melting point of 850 °C on the basis of AN ceramics with $(Sr_{0.7}Bi_{0.2}) HfO_3$. It was found that the sintering temperature was effectively reduced, and all components have high relative densities (>98%), which indicated that BCB significantly promoted the compactness of ceramics during sintering. Moreover, a remarkable $W_{rec}$ of 6.1 $J/cm^3$ and a relatively high $\eta$ of 73% were simultaneously obtained in 0.055SBH-modified AN ceramic with 1 mol% BCB addition under an applied electric field of 330 kV/cm. Pressure-assisted sintering is characterized by the application of external pressure during sintering, which facilitates the mass transport of the grains and, thus, promotes densification. Wu et al. [32] prepared $Ba_{0.3}Sr_{0.7}TiO_3$ ceramic samples by means of spark plasma sintering and observed lower porosity and fewer defects in the SPS samples, which greatly enhanced the breakdown strength. Fang et al. [33] adopted hot pressing sintering to increase the breakdown strength of $TiO_2$-$SiO_2$-$Al_2O_3$-based ceramics to 77.5 kV/cm, which is 1.8 times that of traditional sintered samples. It is found that the increase in density is an important reason for the increase in breakdown strength. However, the unknown dosage of sintering additives limits its application to a certain extent. Although methods such as spark plasma technology and hot-press sintering can obtain products which possess high density, their production costs are relatively high. From the perspective of improving the density of ceramics, this experiment designs a unique microstructure with uneven grains. Through the regulation of the synthesis process, it is expected to achieve the purpose of filling the gaps with grains to increase the degree of densification.

In this study, $AgNbO_3$:$xCeO_2$ ceramic samples were synthesized by the conventional solid-state reaction method, and enhanced AFE stability was obtained by reducing the tolerance factor. $Ce^{4+}$ (r = 1.14 Å, CN = 12) has a smaller radius than $Ag^+$ (r = 1.48 Å, CN = 12) [34], and the lower $t$ will be obtained after the substitution process, as well as a more stable AFE phase. Then, the silver vacancy generated by the substitution process could enlarge the polarization. A new strategy to increase the density of ceramic samples was proposed, that is, by adjusting the grinding time to achieve the heterogeneous grain size. Small-sized grains are used to fill in the gaps between large-sized grains. The effective increase in density contributes to the enhancement of $E_b$. In this paper, the phase structure, microstructure, dielectric properties, and energy storage properties of AN modified by excess $CeO_2$ were measured and discussed.

## 2. Experimental Section

### 2.1. Materials Preparation

The $AgNbO_3$:$xCeO_2$ ($x$ = 0, 1, 2 mol%, abbreviated as ANCe$x$: ANCe0, ANCe1, ANCe2, respectively) ceramics were synthesized by conventional solid-state reaction method. $Ag_2O$ (99.7%, Shanghai Aladdin Biochemical Technology Co., Ltd.), $Nb_2O_5$ (99.99%, Shanghai Aladdin Biochemical Technology Co., Ltd., Shanghai, China), and $CeO_2$ (99.99%, Shanghai Aladdin Biochemical Technology Co., Ltd., Shanghai, China) were used as raw materials,

and they were mixed by planetary ball milling technology. For pure substances, $Ag_2O$ and $Nb_2O_5$ raw materials were first weighed according to half of the stoichiometric ratio, and they were ball milled at 300 rpm for 12 h with absolute ethanol and zirconia balls as the medium. Subsequently, the other half of the stoichiometric ratio of oxide raw materials were added to continue the 12-h ball milling process at 300 rpm. The doped components, such as the doped oxide $CeO_2$, were introduced and ball milled for 12 h at 300 rpm in the beginning, and other oxides were weighed and added in the same way as pure substances. Then, the well-mixed and dried powders were calcined at 900 °C for 6 h in an oxygen environment. After grinding and refining, the powders were subjected to the same process of secondary ball milling. The samples obtained by ball milling were dried, mixed with a 5% mass fraction of polyvinyl alcohol and pressed into pellets with a diameter of 8 mm. A series of ceramic samples were obtained by sintering at 1090–1140 °C for 6 h in an oxygen environment.

### 2.2. Characterization

The density of bulk ceramics was measured by the Archimedes drainage method. The AX124ZH electronic scale and its density measuring module produced by the American Ohaus Company (Parsippany, NJ, USA)were used to measure the density. The phase structure of the ceramic powders was characterized by an X-ray diffractometer (XRD, D8 Advance A25, Bruker, Saarbruken, Germany) with monochromatic Cu K$\alpha$ radiation ($\lambda$ = 1.5405 Å). The XRD patterns were obtained in the $2\theta$ range from 20° to 70°. The surface morphology of bulk ceramics was observed via field emission scanning electron microscopy (SEM, JSM-7500F, JEOL LTD, Tokyo, Japan). It is worth noting that the ceramic samples need to be ground, polished, and thermally etched for 40 min at a temperature below the sintering temperature before testing. Energy Dispersive Spectrometer (EDS, PHENOM PROX, Eindhoven, The Netherlands) is used in conjunction with SEM, and it was used to analyze the element type in the SEM topography of ceramic samples. The grain size distribution of the ceramic samples was determined by the software Nano Measurer (version 1.2.5) based on the measured electron microscope images. The dielectric properties of the ceramics were measured by LCR automatic tester (TH2827A; Changzhou Tong hui Electronics Co., Ltd. Changzhou, China) and dielectric test systems (DPTS20005P1; Yanhe Technology Co., Ltd. Wuhan, China) in the temperature range of 20–450 °C, with the frequency at 10 kHz. It should be noted that both the front and back sides of the samples used for characterization need to be coated with silver electrodes. The energy storage performance of the samples was characterized by the ferroelectric tester. The polarization-electric field loops of the ceramic samples were measured at 10 Hz in silicone oil by a ferroelectric testing system (Radiant Technologies, Albuquerque, NM, USA). The ceramic samples for testing needed to be coated with symmetrical silver electrodes and ground to about 0.1 mm in thickness.

### 3. Results and Discussion

The XRD patterns of the ANCe$x$ are presented in Figure 1a. The details of the relevant peaks are highlighted in the magnified section, as shown in Figure 1b. After comparing with the standard PDF#70-4738, it can be seen that all components possess a pure perovskite structure. It means that $Ce^{4+}$ diffuses successfully into the $AgNbO_3$ lattice. According to the enlarged pictures, the (020), (114), and (220) diffraction peaks moved to higher angles with increasing doping amounts. The shift of the diffraction peak to the right roots in the fact that $Ce^{4+}$ with a smaller radius replaces $Ag^+$ to make the crystal lattice shrink. Since the ionic radius of $Ce^{4+}$ (r = 1.14 Å, CN = 12) is much lower than $Ag^+$ (r = 1.48 Å, CN = 12) at the A-site, conversely, the ionic radius of $Ce^{4+}$ at another coordination number (r = 0.87 Å, CN = 6) is much larger than $Nb^{5+}$ (r= 0.64 Å, CN = 6). Thus, when a solid solution is formed, the lattice volume is reduced by an A-site substitution.

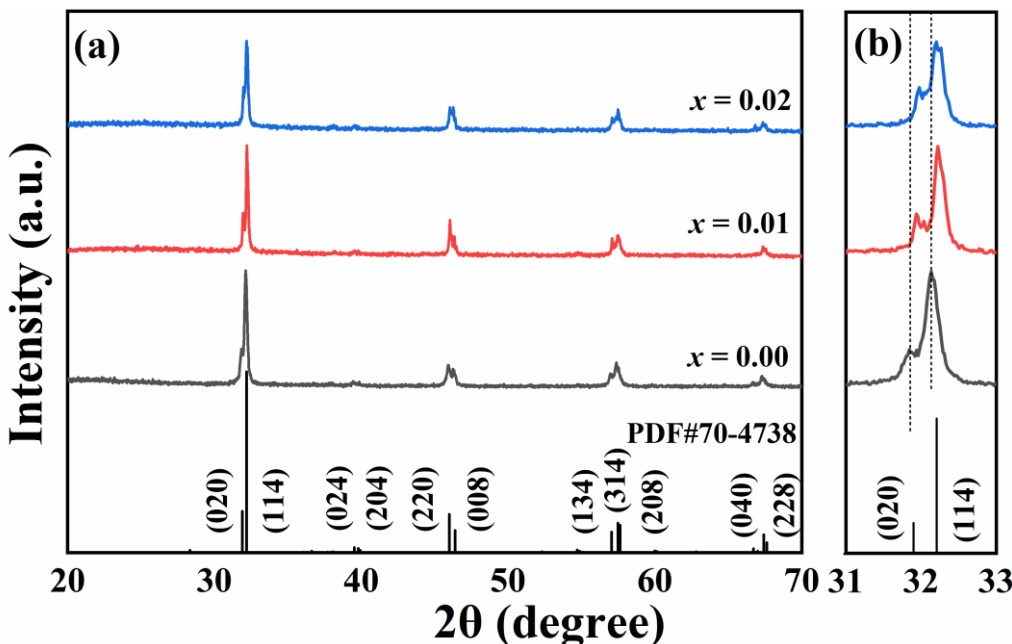

**Figure 1.** (**a**) XRD patterns, (**b**) the enlarged part of diffraction peaks around 32° of ANCe*x* ceramics.

Figure 2a–c presents the SEM images of the ceramics. It can be found that all samples have the characteristics of grain size heterogeneity. The coexistence of large-sized and small-sized grains is attributed to the particularity of the synthesis process. Due to the different milling sequences, for ANCe0, half of the $Ag_2O$ and $Nb_2O_5$ were milled for 24 h, and the other half of the raw material was only milled for 12 h. While for ANCe1 and ANCe2, all the $CeO_2$ were ball-milled for 24 h, and the rest were the same as ANCe0. The special ball milling method affects the degree of mixing of raw materials, resulting in differences in grain size. Figure 2e–h shows the molar percentage of each substance when Nb is used as a reference based on EDS results. For ANCe0, it can be observed in Figure 2e–f that the large grains and small grains contain the same elements, which demonstrates the successful synthesis of silver niobate. Figure 2g–h displays that both sites contain Ag, Nb, and Ce elements. EDS results show that $Ce^{4+}$ was successfully incorporated into the $AgNbO_3$ lattice. The grain size distribution was measured using the software Nano Measurer and displayed in Figure 3. For the purpose of better evaluating the variation trend of small and large-size grains, the analysis of the grain size distribution was carried out for grains above 10 μm and below 10 μm, respectively. The grain size distribution of the small particles shown in Figure 3a–f corresponds to that of the large grains. For the small grains of the ANCe0 component, the value of the most probable grain size (MPGS) according to Gauss fitting is 2.5 μm. Clearly, after $Ce^{4+}$ doping, the MPGS of small grains increased from 2.5 μm in ANCe0 to 4.5 μm in ANCe2, while the MPGS of large grains remained basically unchanged. These results indicate that $Ce^{4+}$ can promote the growth of small grains to a certain extent. In addition, the growth of these small grains has a squeezing effect on the large grains, which inhibits the abnormal growth of the large grains, contributing to the constant profile of large grains. The XRD results show that with the introduction of $CeO_2$, $Ce^{4+}$ replaced $Ag^+$ at the A-site. The substitution of $Ce^{4+}$ for $Ag^+$ resulted in the formation of A-site vacancies. These A-site vacancies can promote the diffusion of ions and the transport of substances during the sintering process and therefore promote the growth of grains. This can well explain the results of grain size distribution. The actual density of the ceramics was determined by the Archimedes method according to formula (5).

$$\rho = \frac{M_1}{M_2 - M_1}(\rho_0 - \rho_L) + \rho_L \tag{5}$$

where $\rho$ is the density of the sample to be measured, $M_1$ and $M_2$ are the measured mass of the sample in air and distilled water, $\rho_0$ is the density of distilled water ($1 \text{ g/cm}^3$), and $\rho_L$ is the density of air ($0.0012 \text{ g/cm}^3$). As shown in Figure 2d, the densities of ANCe$x$ ceramics for $x = 0, 1, 2$ mol% are 6.51, 6.57, and 6.39 g/cm$^3$, respectively. It is clear that the ANCe1 component has the highest density value. In this component, the sizes of large and small particles match well, and the small particles have filled the gaps between the large particles as much as possible. The compensation of the holes makes the density significantly enhanced. Interestingly, a more homogeneous grain distribution was obtained in the composition of ANCe2, along with a reduced actual density. In virtue of the growth of small particles, there were not enough small particles to fill the gaps between large particles, thus accounting for the decreased densification. Weibull distribution was adopted to define the $E_b$ values of the ANCe$x$ ceramics, as shown in Figure 4. In order to ensure the accuracy of the data, it is usually necessary to prepare at least 8-10 samples for breakdown measurement. It can be seen that all components show a linear relationship with a large $\beta$ value, indicating the validity of the Weibull distribution [5,35]. $E_b$ increased to 390 kV/cm when the doping amount was 1 mol%, which is 1.7 times that of the ANCe0 component. When the doping amount reached 2 mol%, $E_b$ was almost consistent with ANCe0. $E_b$ is considered a key parameter to measure energy storage performance. In this research, the composition of ANCe1 achieved the highest degree of densification, which is one of the crucial factors endowing the composition with a relatively high breakdown strength of 390 kV/cm. While for the ANCe2 component, the decrease of $E_b$ is due to the increase of porosity. In the ANCe2 component, due to the growth of small particles, there were not enough small particles to fill the gaps between the large particles. This resulted in an increase in porosity.

For exploring the effect of $CeO_2$ addition on the temperature of the phase transition of ANCe$x$ ceramics, Figure 5a–c depicts the dielectric constant ($\varepsilon_r$) and dielectric loss (tan$\delta$) of all samples at 10 kHz as a function of temperature over the range of 20–450 °C. The anomalous dielectric peaks in the trend chart are caused by the transformation of the phase structure of the ceramic components during the heating process. The four dielectric anomalies corresponding to ANCe0 are related to the phase transitions of $M_1$-$M_2$, $M_2$-$M_3$, $M_3$-O, and O-T, respectively [36]. After doping $Ce^{4+}$, the individual phase transition temperature obviously shifted to a lower temperature. For a detailed analysis of the change in phase transition temperature, the phase diagram of each component was drawn according to the corresponding values in the figure, as shown in Figure 5d. It is clear that with the increase of Ce content, $T_{M1-M2}$ decreased from 71 °C of ANCe0 to 38 °C of ANCe1, and the $M_2$ phase of ANCe2 existed at room temperature, indicating the enhanced stability of the antiferroelectric phase at lower temperatures. $T_{M2-M3}$ decreased gradually, while $T_{M3-O}$ and $T_{O-T}$ remained basically unchanged, which manifests that the ceramic components exhibit antiferroelectricity in a wider temperature range. The enhancement of antiferroelectricity originates from the lower tolerance factor value. When $t < 1$, the AFE phase is more stable. According to formula (4), the substitution of small-radius $Ce^{4+}$ (r = 1.14 Å, CN = 12) for large-radius $Ag^+$ (r = 1.48 Å, CN = 12) can effectively reduce the tolerance factor, thereby enhancing the antiferroelectricity of ceramic components. It can be observed from Figure 5a–c that the dielectric peaks corresponding to $T_{M2-M3}$ gradually widened, and a diffuse phase transition occurred, indicating enhanced relaxation characteristics. Moreover, all samples exhibited a low dielectric loss in the range from room temperature to 450 °C. This is thought to be associated with the high electrical insulation, which is conducive to achieving high $E_b$.

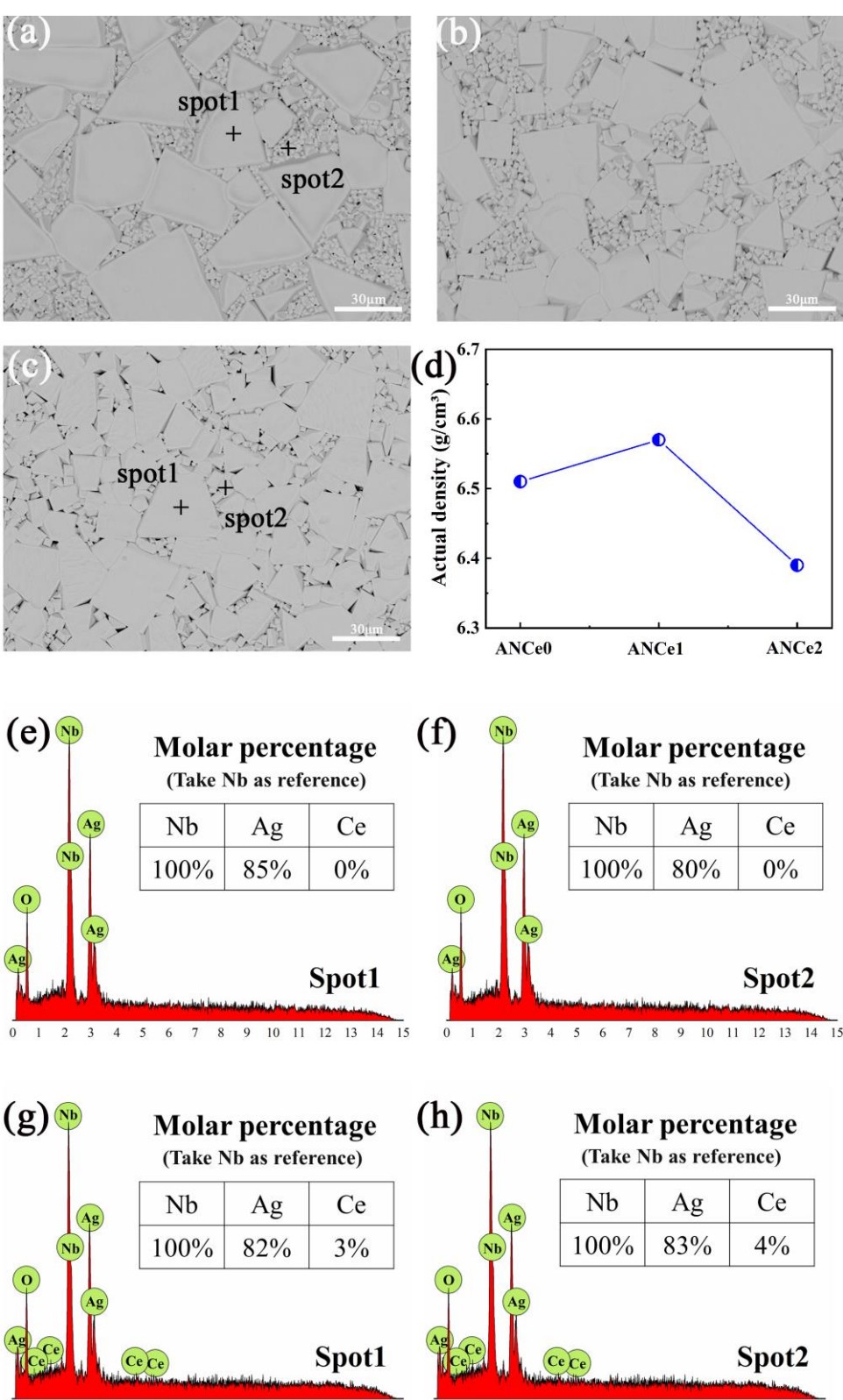

**Figure 2.** SEM images, the actual density, and EDS results of ANCe*x* ceramics. SEM images of (**a**) ANCe0, (**b**) ANCe1 and (**c**) ANCe2 ceramics, and (**d**) The actual density of ANCex ceramics. EDS results of (**e**,**f**) ANCe0 and (**g**,**h**) ANCe2 ceramics.

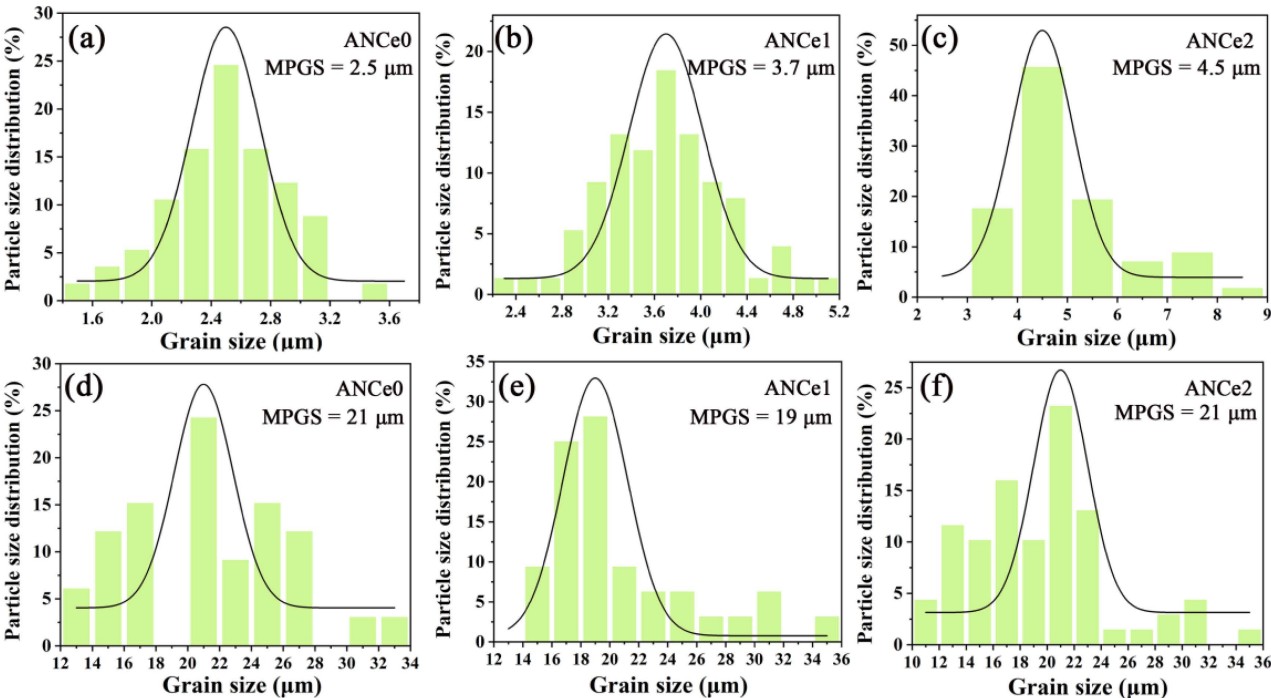

**Figure 3.** Grain size distribution of ANCe*x* ceramics. Grain size distribution of small particles of (**a**) ANCe0, (**b**) ANCe1, and (**c**) ANCe2 ceramics. Grain size distribution of big particles of (**d**) ANCe0, (**e**) ANCe1, and (**f**) ANCe2 ceramics.

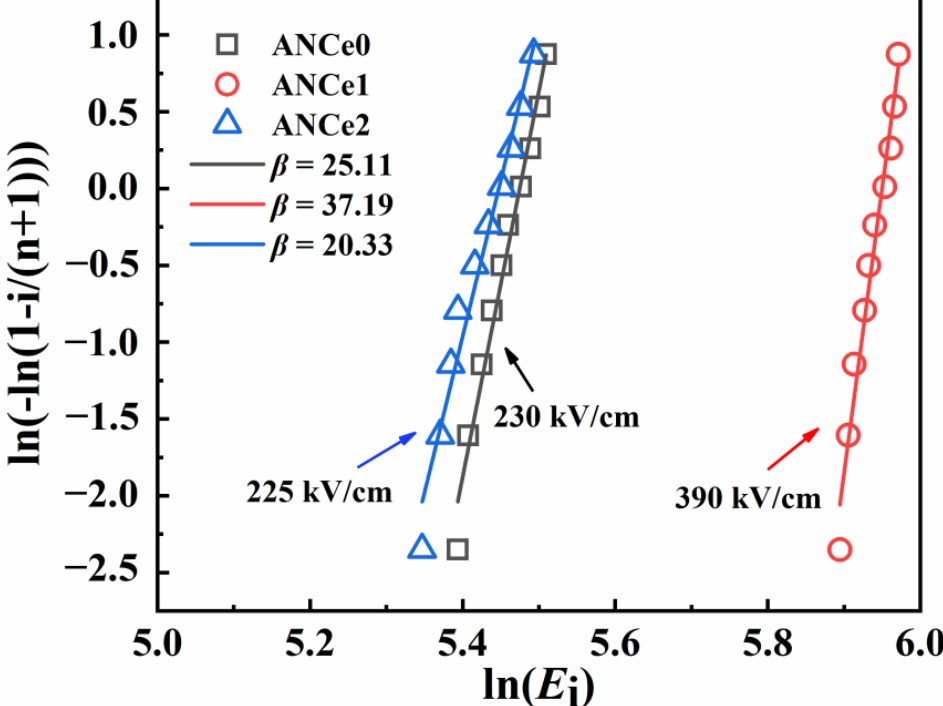

**Figure 4.** Weibull distribution of dielectric breakdown strength ($E_b$) for ANCe*x* ceramics.

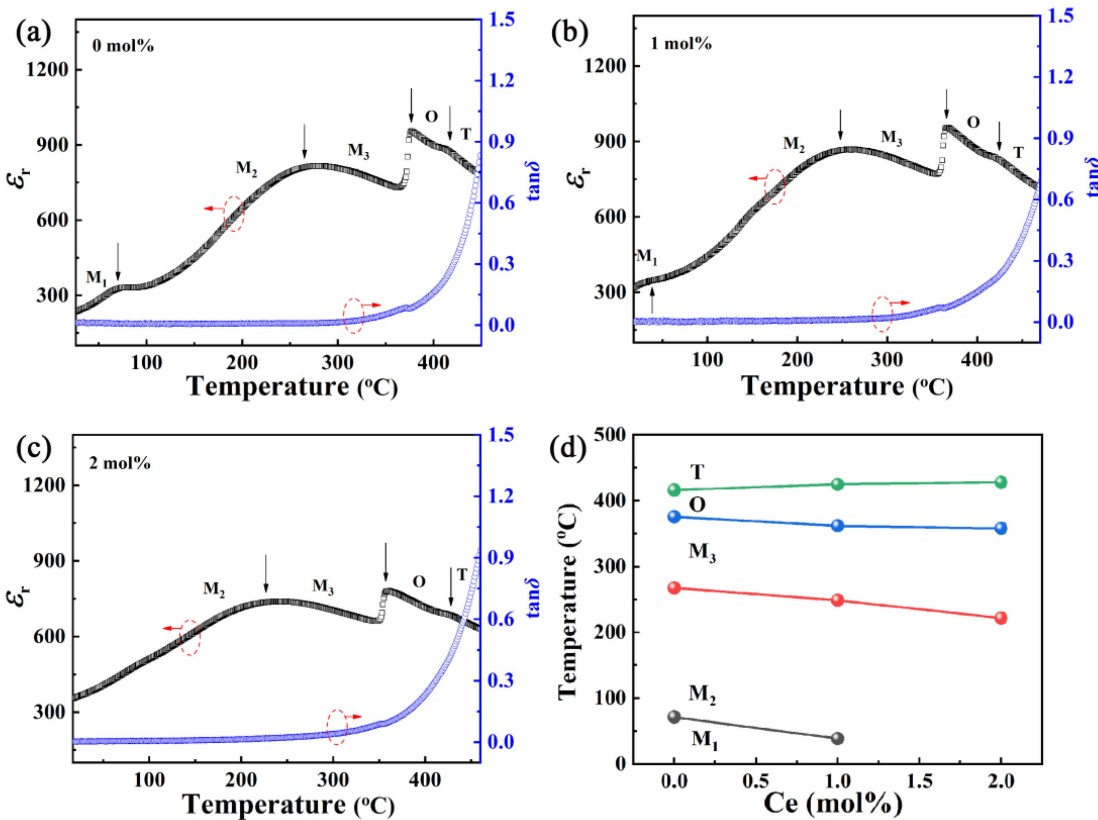

**Figure 5.** Temperature dependence of the dielectric constant ($\varepsilon_r$) and dielectric loss (tan$\delta$) of (**a**)ANCe0, (**b**) ANCe1, and (**c**) ANCe2 ceramics. (**d**) Phase diagram of the ANCe*x* ceramics. (The red circles and arrows point to the axes corresponding to the data and black arrows point to the phase transition temperature).

　　To explore the effect of the introduction of $Ce^{4+}$ on the energy storage performance, the unipolar polarization loops of ANCe*x* ceramics under their respective maximum tolerable electric fields were tested and displayed in Figure 6a. The results demonstrate that when the doping content was greater than or equal to 1 mol%, the ceramic samples exhibited typical double hysteresis loop characteristics. The *P-E* loops of the ceramic samples became narrower after doping $Ce^{4+}$, which is beneficial for obtaining better energy storage performance. The relevant energy storage data, such as polarization, phase transition electric field, and energy storage density determined according to the *P-E* loops of the ANCe*x* ceramic composition, are listed in Table 1. With the introduction of $Ce^{4+}$, $P_{max}$ first increased and then decreased. The high $P_{max}$ obtained in the ANCe1 component is correlated with the achievement of high $E_b$. Meanwhile, the presence of A-site vacancies can also contribute to a high $P_{max}$ value. Numerous studies have found that polarization intensity is largely affected by cation vacancies, and the essence is that the existence of cation vacancies facilitates the steering of electric dipoles [22,37,38]. The reduction in $P_{max}$ of the ANCe2 component is because its $E_b$ is much lower than ANCe1—limiting the phase transition of the component from the AFE phase to the FE phase. For the ANCe2 ceramic, its lower density, compared to ANCe1, results in a smaller $E_b$. The $P_r$ of ceramic samples decreased monotonically with the increasing doping amount, which is attributed to the increasing relaxation behavior. The enhanced relaxation characteristics are reflected in the gradually widening dielectric peaks. This has been discussed in the dielectric properties section. A smaller $P_r$ is effective in reducing the hysteresis loss of the ceramic samples, leading to a higher $\eta$. The phase transition electric field is also an important index used to measure the stability of antiferroelectricity. With the introduction of $Ce^{4+}$, $E_F$ displayed an overall downward trend. This result implies that the phase transition from the antiferroelectric

phase to the ferroelectric phase can be induced at a lower electric field. In contrast, the monotonically increasing trend of $E_A$ reflects the small hysteresis from the FE phase to the AFE phase with $Ce^{4+}$ doping. The increase of $E_A$ indicates enhanced AFE stability, which is achieved by the decrease of t after the incorporation of $Ce^{4+}$. These states reduce the $\Delta E$ ($E_A - E_F$) value significantly by upshifting $E_A$ and downshifting $E_F$ simultaneously. With the increase of the $Ce^{4+}$ doping amount, $\Delta E$ decreased gradually and promoted the acquisition of high $\eta$. Combined with low $P_r$ and small $\Delta E$, the ANCe2 component has a slimmer hysteresis loop and, thus, a higher $\eta$. Figure 6d shows the trend of energy storage density and energy storage efficiency calculated from the *P-E* loops. For ANCe1 ceramic, with the substitution of small-radius $Ce^{4+}$ (r = 1.14 Å, CN = 12) for large-radius $Ag^+$ (r = 1.48 Å, CN = 12), the average radius of the A-site ion is significantly reduced compared with ANCe0. This suggests that the tolerance factor *t* is reduced and the antiferroelectricity is enhanced. As can be seen, a superior $W_{rec}$ of 5.04 J/cm$^3$ and a $\eta$ of 46.2% are obtained under an applied electric field of 390 kV/cm for ANCe1 ceramics, in virtue of the perfect combination of the $P_{max}$, $E_b$, and $E_A$. The energy storage density of this component is 1.9 times higher than that of ANCe0, suggesting that the introduction of $Ce^{4+}$ is conducive to optimizing energy storage performance. For the ANCe2 component, its $P_{max}$ is limited by the smaller $E_b$, thus exhibiting a relatively low $W_{rec}$. However, its smaller $P_r$ and lower $\Delta E$ make it achieve a higher $\eta$ of 55.4%. In general, the introduction of $CeO_2$ benefits lower *t* value and $P_r$, as well as a more stable AFE phase. Samples show different performances depending on the amount added. A small amount of $CeO_2$-doping is conducive to obtaining higher $P_{max}$ and $E_b$, while a high concentration of $CeO_2$-doping is prone to decrease $P_r$ and $\Delta E$. This affords a step forward for dielectric materials with comprehensive energy storage properties.

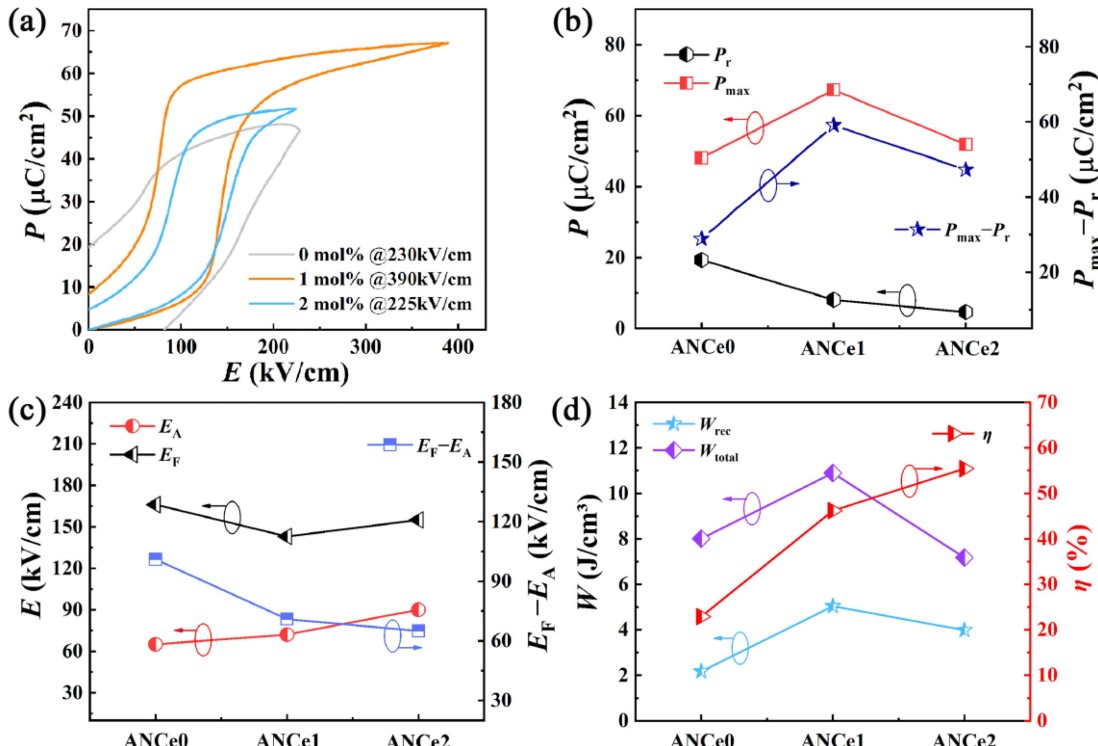

**Figure 6.** Energy storage properties of ANCe*x* ceramics. (**a**) Unipolar *P-E* loops, (**b**) $P_{max}$, $P_r$ and $P_{max} - P_r$, (The red and black circles and arrows point to the *P* axis, and the blue circles and arrows point to the $P_{max} - P_r$ axis.) (**c**) $E_A$, $E_F$ and $E_A - E_F$, (The red and black circles and arrows point to the *E* axis, and the blue circles and arrows point to the $E_F - E_A$ axis.) and (**d**) $W_{total}$, $W_{rec}$ and $\eta$. ((The purple and blue circles and arrows point to the *W* axis, and the red circles and arrows point to the $\eta$ axis.)

**Table 1.** Parameters related to energy storage properties of ANCe*x* samples.

| Samples | $P_m$ ($\mu$C/cm$^2$) | $P_r$ ($\mu$C/cm$^2$) | $E_F$ (kV/cm) | $E_A$ (kV/cm) | $\Delta E$ (kV/cm) | $W_{rec}$ (J/cm$^3$) | $\eta$ (%) |
|---|---|---|---|---|---|---|---|
| ANCe0 | 48.1 | 19.3 | 166 | 65 | 101 | 1.73 | 22.9 |
| ANCe1 | 67.2 | 8.1 | 143 | 72 | 71 | 5.04 | 46.2 |
| ANCe2 | 51.9 | 4.6 | 155 | 90 | 65 | 3.98 | 55.4 |

## 4. Conclusions

In this research, the $W_{rec}$ of AgNbO$_3$:$x$CeO$_2$ ceramics was improved in the following four aspects. It consists of enhancing AFE stability by reducing $t$, reducing $P_r$ by enhancing relaxation characteristics, increasing $P_{max}$ by introducing A-site vacancy, and enlarging the $E_b$ by achieving a higher density of ceramics. The phase transition temperature $T_{M1\text{-}M2}$ gradually moved to a lower temperature, which further confirms the enhanced AFE stability. Moreover, the gradual widening of dielectric peaks verifies the enhancement of relaxation behavior. Additionally, the enhancement of $E_b$ up to 390 kV/cm is derived from the increase in density, which is essentially due to the filling effect of small grains on the gaps between large grains. The ANCe*x* ceramics were successfully fabricated by the conventional solid-state reaction method. This effect is achieved by the regulation of CeO$_2$ amount and the grinding procedure, and thus a higher degree of densification is realized in ANCe1 ceramic. Therefore, a high $W_{rec}$ of 5.04 J/cm$^3$ and a $\eta$ of 46.2% were achieved in the ANCe1 component owing to the combined effect of the increased $P_{max}$, $E_A$, and $E_b$. Moreover, a relatively high $\eta$ of 55.4% that was obtained in ANCe2 composition attributed to the lower $P_r$ and smaller $\Delta E$. These results indicate that the components with different doping amounts have outstanding properties in different fields and provide guidance for us to search for ceramic materials with comprehensive energy storage properties.

**Author Contributions:** All authors contributed to the conception and design of the study. Data curation, F.H.; Formal analysis, W.W.; Methodology, T.F.; Resources, G.L.; Supervision, J.W.; Writing—original draft, K.A. All authors have read and agreed to the published version of the manuscript.

**Funding:** This work is funded by the National Natural Science Foundation of China (Grant No.51302061), the Natural Science Foundation of Hebei province (Grant No.E2014201076 and E2020201021), and the Research Innovation Team of the College of Chemistry and Environmental Science of Hebei University (Grant No. hxkytd2102).

**Institutional Review Board Statement:** Not applicable.

**Informed Consent Statement:** Not applicable.

**Data Availability Statement:** Not applicable.

**Conflicts of Interest:** The authors declare no conflict of interest.

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
