# Peer review of "Enhanced Energy Storage Performance of AgNbO3:xCeO2 by Synergistic Strategies of Tolerance Factor and Density Regulations"

_coatings, doi:10.3390/coatings13030534_

Round 1

Reviewer 1 Report (Previous Reviewer 1)

The authors study the Ce-doping of AgNbO3 ceramics for energy storage. It is of general interest due to the energy storage perspective, however, I have some comments.

First major comment, why is the Ce-doping of interest? Two key parameters Eb and Pmax are controlled by the higher density of Ce1. Does the change in  ΔE and η make the compositional effect sufficient to prove an influence of Ce-doping? This should be clearly stated in the manuscript.

Second major comment, I do not think the study fits the aims and scope of Coatings. I suggest the authors to revise based on my comments and transfer to, e.g., Ceramics that seems like a better option (scope: ceramics application in energy).

Line 185 refers to Fig. 1c, but that figure does not exist.

In line 186-187, the austhors state "all components posess a pure perovskite phase". However, it seems small XRD peaks are visible around 38 and 40 degree - what is the origin of these? Also, I suggest to perform Rietveld refinement to obtain additional info on the crystal structural parameters.

In figure 2, why was the EDS of Ce1 not provided? Also, I suggest to change the atomic percentage of the elements to mol% to compare the actual to the theoretical contents.

In Fig. 5d, what happens to the M1 phase? If is does not exist at 2%Ce, I suggest to add it as "0".

Author Response

Reviewer 2 Report (New Reviewer)

I find that the manuscript from the title: "Enhanced energy storage performance of AgNbO3:xCeO2 by synergistic strategies of tolerance factor and density regulations" is very interesting paper.

The authors developed an AgNbO3:xCeO2 (x=0, 1, 2 mol%) ceramics were prepared by the conventional solid-state reaction method, in order to the optimization of energy storage properties. Autori navode de je the enhanced antiferroelectric (AFE) stability and the increased breakdown strength (Eb), due to the increase of actual density, which is achieved through the regulation of CeO2 amount and grinding procedure in the experimental process.

However, the Figure 2 presents the EDS results, which show that the presence of Ce4+/3+ ions is in a very small percentage (0.36% and 0.49%), which is much lower than the planned value (1% and 2 mol%) before the synthesis procedure.

I ask the authors to explain the mentioned improvements of antiferroelectric (AFE) stability and the breakdown strength (Eb), considering that the obtained AgNbO3:xCeO2 ceramics did not incorporate Ce4+/3+ ions in the planned amount.

I recommend accept after minor revision (corrections to minor methodological errors and text editing).

Author Response

Reviewer 3 Report (New Reviewer)

In this work, AgNbO3:xCeO2 materials were prepared and investigated for improved features. The presented results might be attractive for the specialists in advanced dielectrics with increased energy storage performance.

 1. Sintering was performed between 1090 and 1140 oC. However, it is not clear which samples were sintered at what temperature. Could the authors please explain what was the effect of the sintering temperature ?

 2. Are the traces in Fig. 1 depending on x (Ce quantity) ?. Please clearly specify the content for each trace. In addition, Fig. 1c is mentioned at page 5, but there is no Fig. 1c.

 3. There are two EDS graphs below Fig. 2e and Fig. 2f. Please add legends also for these figures.

Round 2

Reviewer 1 Report (Previous Reviewer 1)

The authors argued well for their choices. I only have one minor comment before the paper can be accepted.

With the newly added composition in mol% from EDS measurements, I follow the argument in the response letter about the difference in Ce content compared to the nominal one. However, why is the total of Ce+Ag content much smaller than that of Nb? Cerium is added in excess to the AgNbO3 composition, thus, Ag+Ce should be larger than Nb. Is it expected to observe an A-site vacancy as well? If that is the case, what happened to the Ag that is missing?

Author Response

This manuscript is a resubmission of an earlier submission. The following is a list of the peer review reports and author responses from that submission.

Round 1

Reviewer 1 Report

The manuscript concerns energy storage of perovskite ceramics. In general, the topic is of great interest, however, I have a major concern about the study.

The authors study the impact of CeO2 doping of AgNbO3, however, the energy storage properties are related to the breakdown strength that it further related to the density. As the density varies among the samples, it seems that is the largest influence, and the CeO2 doping is not well explored. Could the authors please clarify why the CeO2 doping is of interest and how this is shown by the results given the density change in the ceramics?

The last paragraph of the introduction (lines 101-114) should be shortened and only discuss what is to be investigated in the paper and not present results.

I do have some further comments to the manuscript found below:

1. As CeO2 is added in excess to the AgNbO3 composition, it should be discuss and further clarified e.g. by additional experiments how this occurs. Will Ag or Ce segregate as oxides in addition to a Ce-doped AgNbO3 structure, or does the perovskite phase have a B-site vacancy as the A-site will be in excess?

2. In line 158-160 it should be clarified that Ce will substitute Ag and nog Nb beccause of the coordination of Ce4+ (12-fold), as the B-site needs to be 6-fold coordinated.

3. For Figure 2, please show where Spot 1 and Spot 2 EDS are measured. Also, the d)-e)-f) should be updated in the figure caption so it fits the actual images.

Reviewer 2 Report

REFEREE REPORT

on paper Enhanced energy storage performance of AgNbO3:xCeO2 by

synergistic strategies of tolerance factor and densities regulations

by authors Ke An, Gang Li, Tingting Fan, Feng Huang, Wenlin Wang and Jing Wang,

submitted to Coatings

The paper Enhanced energy storage performance of AgNbO3:xCeO2 by synergistic strategies of tolerance factor and densities regulations is devoted to preparation and investigation of the AgNbO3:xCeO2 (x=0, 1, 2 mol%) ceramics. Conventional solid-state reaction method was used for the samples obtaining. XRD, SEM techniques and LCR, ferroelectric testing system measurements were applied for the samples characterization. The data are reliable and do not cause much doubt. The topic of this paper is critically actual especially in the field of the energy storage application. Nevertheless, there are several points before the paper can be published. I hope that authors after major revisions can improve the paper and can publish it in Coatings.

1.     The Introduction part is too short. It must be improved with the new literature about ceramics and I suggest using the following reference (see and discuss:

https://doi.org/10.1016/j.jallcom.2022.164577; https://doi.org/j.ceramint.2019.05.039).

2.     Why did the paper were submitted to the Coatings since the paper about bulk ceramics samples?

3.     Did you evaluate some XRD peaks shift (Fig. 1)? If yes, why so?

4.     The resolution of all figures is poor. It should be improved.

5.     The distribution of the average grain size should be added to the manuscript. Please use and discuss simple method described in [https://doi.org/10.3390/nano12101642].

6.     How did you evaluate the relative density of the samples?

7.     How do you explain the behavior of the unipolar P-E loops?

8.     The Conclusion part is too short, please improve it.

9.     There are some insufficient typos and English mistakes in the text.

But any way I impressed by this paper. But authors must explain some details and improve the paper in accordance with my comments. The paper should be sent to me for the second analysis after the major revisions.

Reviewer 3 Report

The manuscript entitled: “Enhanced energy storage performance of AgNbO3:xCeO2 by synergistic strategies of tolerance factor and densities regulations” submitted to Coatings as Article  describes synthesis and characterization of new ceramic materials based on AgNbO3 doped with CeO2. For preparation of samples authors used simple conventional solid-state reaction. Afterwards all samples were characterised using typical standard methods such as: XRD, SEM, dielectric constant and unipolar P-E loops. It was noticed that the addition of the CeO2 change the morphology of the morphological structure resulting in increase of possible energy storage efficiency however the value of the maximum power dropped.

The topic presented in the manuscript bring into light some interesting finding regarding the strategy of ceramic materials composition for energy storage. I do recommend this article for publishing in its current form.

Reviewer 4 Report

It is an interesting paper about the enhanced energy storage performances of AgNbO3:xCeO2. The paper is well organized, but some changes are required prior to publication.

1)       In the caption of Figure 2 there is clearly an error in the e-f sequence. In addition, the spots on the SEM images are not evident and the caption itself is not sufficiently explicative.

2)       In general, all the captions of the figures of the paper should be more explicative.

3)       In the “Materials preparation” section the undoped sample is indicated as ANCe0 while in the caption of Figure 3 and in Table 1 the same sample is indicated as AN.

4)       The English should be revised since some sentences are really difficult to read.

Round 2

Reviewer 1 Report

In lines 185-189 the enlarged XRD figures are discussed regarding change in peak position (Fig 1 b and c). I agree that the 114 plane seems not to change from 1 to 2 mol% CeO2 addition, however, the peak shape changes significantly in the area between the two dashed lines. Also, when the CeO2 content increases from 1 to 2 mol%, the 220 and 008 planes seems to merge in Fig. 1c. Please discuss these two points and how this might be related to the incorporation of CeO2 in the perovskite.

Reviewer 2 Report

Revised version can be accepted
